# Electron transfer in the respiratory chain at low salinity

Ana Paula Lobez[1,7], Fei Wu[1,7], Justin M. Di Trani[2,5], John L. Rubinstein [2,3,4], Mikael Oliveberg[1] ✉, Peter Brzezinski [1] ✉ & Agnes Moe [1,6] ✉

Recent studies have established that cellular electrostatic interactions are more influential than assumed previously. Here, we use cryo-EM and perform steady-state kinetic studies to investigate electrostatic interactions between cytochrome (cyt.) $c$ and the complex (C) III$_2$-IV supercomplex from *Saccharomyces cerevisiae* at low salinity. The kinetic studies show a sharp transition with a Hill coefficient ≥2, which together with the cryo-EM data at 2.4 Å resolution indicate multiple cyt. $c$ molecules bound along the supercomplex surface. Negatively charged loops of CIII$_2$ subunits Qcr6 and Qcr9 become structured to interact with cyt. $c$. In addition, the higher resolution allows us to identify water molecules in proton pathways of CIV and, to the best of our knowledge, previously unresolved cardiolipin molecules. In conclusion, the lowered electrostatic screening renders engagement of multiple cyt. $c$ molecules that are directed by electrostatically structured CIII$_2$ loops to conduct electron transfer between CIII$_2$ and CIV.

The function of a living cell relies on specific interactions between intracellular proteins. In many cases, these interactions are of electrostatic origin and, therefore, dependent on the ionic strength of the intracellular medium. For *Escherichia coli* cells, the intracellular osmotic pressure matches an extracellular solution that contains 150 mM monovalent salt such as KCl[1,2]. Hence, solutions with a salinity equivalent to 150 mM monovalent salt have long been used to mimic the intracellular conditions (e.g[3].,). However, in addition to small ions, the cytoplasm presents a highly crowded environment that contains a wide range of charged macromolecules and organic compounds. A vast majority of the colloidal-sized macromolecules such as nucleic acids, membranes, and proteins carry a net-negative charge[2,4–6]. In addition, charged organic compounds are typically anions (e.g., ADP/ATP, glutamate, acetate). In other words, to maintain electroneutrality in the intracellular medium, the concentration of small anions such as Cl⁻ is much lower than the concentration of cations such as K⁺[1,2]. Because the majority of negative charges are associated with large polyanions, electrostatic interactions between macromolecules within the cell are considerably stronger than in a 150 mM salt solution, yielding a Debye screening length ($\lambda_D$) of ~ 2.2 nm instead of $\lambda_D \cong 0.8$ nm in 150 mM monovalent salt. As shown by Wennerström et al.[2], a realistic mimic of the intracellular electrostatic conditions in *E. coli* is ~ 20 mM monovalent salt[2].

A well-known example where electrostatic interactions are involved in guiding biochemical processes is the respiratory chain of aerobic organisms. In this chain, the electron carrier cytochrome (cyt.) $c$ has a net positive charge and a dipole moment[7,8]. The molecule docks by facing its heme group near one of its positively charged surfaces towards the strongly negatively charged surfaces of its electron donor or acceptor cytochrome $bc_1$ (also known as complex (C) III) or cytochrome $c$ oxidase (also known as CIV), respectively. Hence, the association of cyt. $c$ with its partner proteins is strongly dependent on the solution ionic strength[9].

[1]Department of Biochemistry and Biophysics, The Arrhenius Laboratories for Natural Sciences, Stockholm University, Stockholm, Sweden. [2]Molecular Medicine program, The Hospital for Sick Children, 686 Bay Street, Toronto, Ontario, Canada. [3]Department of Medical Biophysics, The University of Toronto, 101 College Street, Toronto, Ontario, Canada. [4]Department of Biochemistry, The University of Toronto, 1 Kings College Circle, Toronto, Ontario, Canada. [5]Present address: Department of Biochemistry, University of Alberta, Edmonton, AB T6G 2H7, Canada. [6]Present address: Institute of Biochemistry and Molecular Medicine, University of Bern, Bühlstrasse 28, Bern, Switzerland. [7]These authors contributed equally: Ana Paula Lobez, Fei Wu. ✉e-mail: mikael@dbb.su.se; peterb@dbb.su.se; agnes.moe@unibe.ch

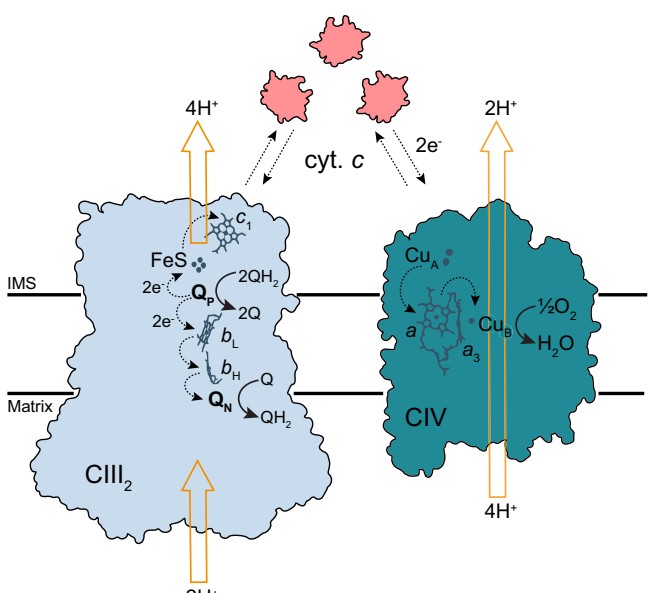

**Fig. 1 | Schematic view of electron and proton transfer by CIII₂ and CIV.** Electron transfer between CIII₂ and CIV is mediated via soluble cyt. *c*. Reactions catalyzed by one-half of the CIII₂ homodimer and CIV are detailed. Quinol and quinone-binding sites are indicated as $Q_P$ and $Q_N$, respectively. Dashed black arrows indicate electron transfer, while yellow arrows indicate proton-transfer reactions. IMS is intermembrane space.

Complex III₂ is an obligate dimer, which receives electrons from membrane-bound ubiquinol (QH₂) (Fig. 1). In each CIII of the CIII₂ dimer, QH₂ binds and is oxidized in the $Q_o$ site (also called $Q_P$), associated with proton release to the positive (*p*) side of the membrane. Upon oxidation, the two electrons are bifurcated; one electron is first transferred to a Rieske iron-sulfur protein and then to cytochrome $c_1$ (cyt. $c_1$), while the second electron is transferred consecutively to hemes $b_L$ and $b_H$, and finally to quinone (Q) bound in the $Q_i$ site (also called $Q_N$). The sequence of reactions is repeated, leading to a reduction of Q in the $Q_i$ site by two electrons, which is associated with proton uptake from the negative (*n*) side of the membrane. The resulting QH₂ dissociates and equilibrates with the QH₂ pool in the membrane (for reviews, see ref. [10–13]). The electron at cyt. $c_1$ is transferred to water-soluble cyt. *c*, which resides in the periplasm of bacteria or in the intermembrane space of mitochondria, i.e., the *p*-side of the membrane (Fig. 1). In CIV, electrons are first transferred to a di-nuclear $Cu_A$ site, located on the *p*-side of CIV, and then consecutively to heme *a* and the heme $a_3$-$Cu_B$ catalytic site where molecular oxygen binds and is reduced to water. Formation of the product water is linked to proton uptake from the *n* side of the membrane. In addition, each electron transfer to O₂ at the catalytic site is linked to the pumping of one proton across the membrane from the *n* to the *p*-side (for review, see ref. [14,15]).

While both CIII₂ and CIV function independently in isolation, recent biochemical and structural studies have shown that they are associated with forming supercomplexes[16–21]. For example, in the yeast *S. cerevisiae,* these supercomplexes are composed of a CIII₂ dimer flanked by one or two copies of CIV (CIII₂CIV₁/₂)[16,22–29]. The formation of a supercomplex composed of CIII₂ and CIV renders an alternative electron-transfer mechanism by 2D diffusion of the positively charged cyt. *c* along the negatively charged surface[20,30–34]. This scenario resembles the direct electron transfer from the FeS center in each CIII in CIII₂ to $Cu_A$ in CIV, via a di-heme cyt. *cc* domain, in the obligate CIII₂CIV₂ supercomplexes from *Mycobacterium smegmatis*[35–37] and *Corynebacterium glutamicum*[38,39].

Results from experimental and theoretical studies indicate that the cyt. *c*-mediated electron transfer between CIII₂ and CIV is rate limiting[30,40], i.e., the QH₂ oxidation:O₂ reduction activity is dependent on electrostatic interactions between cyt. *c* and its partner proteins. In this study, we employ this reaction to study and understand the role of ionic strength on inter-molecular electron transfer. Recent combined structural and kinetic studies performed at 150 mM KCl[30] showed that in *S. cerevisiae,* the negatively charged surface formed collectively by CIII₂ and CIV within the CIII₂CIV₁/₂ supercomplex presents stronger attractive electrostatic binding of cyt. *c* than with CIV or CIII₂ alone. Furthermore, the cryo-EM data showed that in the supercomplex, cyt. *c* is bound at either CIII₂ or CIV or between these complexes in approximately equal proportions. Collectively, the data suggested that cyt. *c* shuttles between the donor and acceptor sites by 2D surface diffusion.

The recent finding that the use of a 150 mM "physiological" ionic strength ($\lambda_D \cong 0.8$ nm) yields much weaker electrostatic interactions than those encountered within the cell encouraged us to investigate the effect of lowering the ionic strength on the mechanism of electron transfer between CIII₂ and CIV. The study of Wennerström et al.[2] is based on a quantitative analysis of *E. coli* cells. The charge distribution in the mitochondrial intermembrane space is likely to be different from that in *E. coli*, but the same principles governing the Debye screening length apply. To investigate the effect of a more realistic, longer Debye screening length[2], we measured the kinetics of electron transfer from QH₂ to O₂ as a function of cyt. *c* concentration and determined the cryo-EM structure of the CIII₂CIV₁/₂-cyt. *c* co-complex at 20 mM monovalent salt concentration ($\lambda_D \cong 2.2$ nm). In contrast to data obtained at 150 mM ionic strength, at low salinity, the kinetic data show a sharp increase in the inter-complex electron-transfer rate at a cyt. *c*:supercomplex ratio of ~10 suggesting cooperative binding of more than one cyt. *c* molecule for each CIII-CIV pair, which is consistent with the cryo-EM data. Furthermore, the cyt. *c*-supercomplex interactions are facilitated by cyt. *c* binding to negatively charged loops of subunits Qcr6 and Qcr9 from each of CIII monomers.

## Results

### Isolation and activity of the supercomplex

The *S. cerevisiae* CIII₂CIV₁/₂ supercomplex was purified using a FLAG tag on subunit Cox6 of CIV[30,41] (Supplementary Fig. S1a, b). Absorption spectroscopy of the preparation yielded different spectra (Supplementary Fig. S1b) consistent with a mixture of CIII₂CIV₁ and CIII₂CIV₂ supercomplexes, as observed previously[28–30,42].

The activity of CIII₂ was measured by monitoring the reduction of cyt. *c* spectrophotometrically at 550 nm, upon addition of reduced quinol to a sample containing cyt. *c* and the CIII₂CIV₁/₂ supercomplex with CIV inhibited by cyanide. At 50 μM cyt. *c* and 20 mM KCl, this activity was 80 ± 6 e⁻/s (mean ± SD, *n* = 4 independent measurements, two distinct samples), which is similar to that measured at 150 mM KCl (90 ± 20 e⁻/s[30]). The activity of CIV was measured by monitoring the O₂-reduction rate using a Clarke electrode upon the addition of the CIII₂CIV₁/₂ supercomplex to a solution containing ascorbate and cyt. *c*. At 50 μM cyt. *c* and 20 mM KCl, this activity was 120 ± 10 e⁻/s (SD, *n* = 4 measurements), which is lower than that measured at 150 mM KCl (450 ± 20 e⁻/s[30]).

The combined activities of CIII₂ and CIV₁/₂ in the supercomplex activity were measured by monitoring the reduction of O₂ upon the addition of reduced quinol (QH₂) in the presence of a fixed supercomplex concentration (Supplementary Fig. S1c), 20 mM KCl, and a variable concentration of cyt. *c* (Fig. 2a). The figure also shows data from the same experiment, published previously[30], but with 150 mM KCl. No activity was observed prior to the addition of cyt. *c*. Upon increasing the cyt. *c* concentration, the supercomplex activity increased, but up to ~ 0.15 μM cyt. *c* (cyt. *c*:supercomplex ratio of ~ 7.5), it remained at a lower level than the activity measured at the same

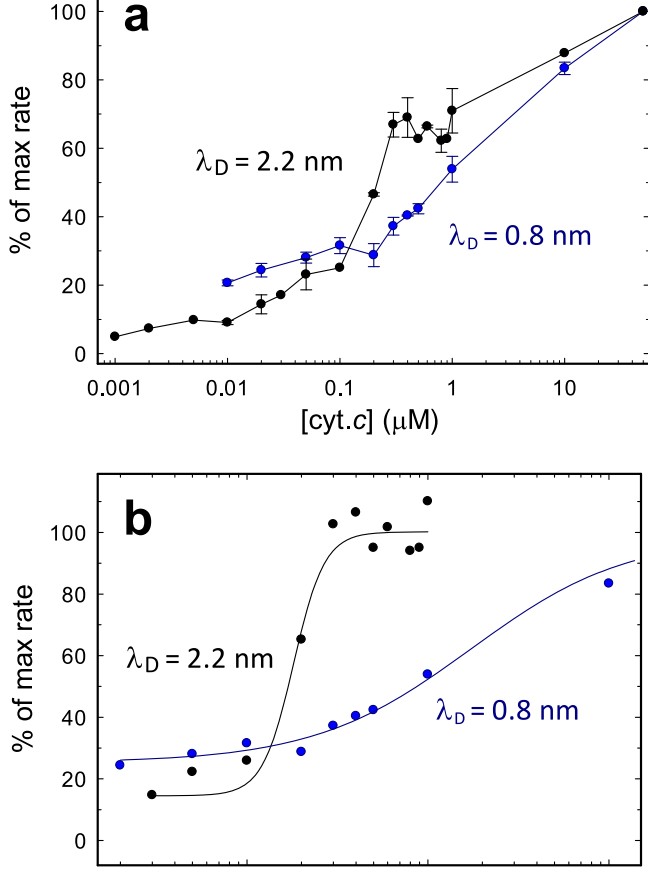

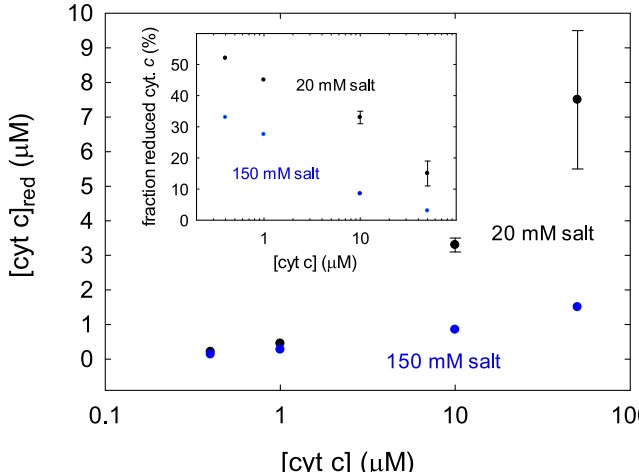

**Fig. 3 | Concentration and fraction reduced cyt. *c*.** The concentration and fraction (inset) reduced cyt. *c* as a function of cyt. *c* concentration for 20 mM (black) and 150 mM (blue) monovalent salt concentrations were measured during the turnover of the supercomplex. Data are average of two technical replicates (one preparation was used because the experiment is focused on the effects of changes in ionic strength). Error bars are calculated as standard deviation.

Fig. 2b shows a fit with a Hill equation to the data at 150 mM KCl[30] with $n = 1$, yielding $K_A = 1.7\ \mu M$.

### Fraction reduced cyt. *c* at steady state

We next calculated the fraction reduced cyt. *c* in solution during the turnover of the supercomplex, i.e., upon addition of QH₂ (Fig. 3). Measurements were performed at cyt. *c* concentrations $\geq 0.4\ \mu M$ where cyt. *c* absorbance could be quantified. Above 0.4 μM cyt. *c*, i.e., at a cyt. *c*:supercomplex ratio of > 20, the concentration of reduced cyt. *c* increased with increasing cyt. *c* concentration both at 20 mM and at 150 mM KCl, but the rate of increase was greater at 20 mM than at 150 mM salt. The observed behavior is explained by electrostatic interactions between cyt. *c* and the supercomplex as outlined in detail in the Discussion section.

### Structure determination by cryo-EM

We mixed the supercomplex and cyt. *c* at a ~1:12 ratio (~10 μM supercomplex) in a solution with an ionic strength of 20 mM and applied the sample to holey gold grids for analysis by cryo-EM (Supplementary Table S1 and Supplementary Fig. S2). Analysis of the resulting three-dimensional (3D) maps showed a CIII₂ dimer flanked by either one or two monomers of CIV, consistent with earlier observations[28–30,42,43]. Combining the CIII₂CIV₂ and CIII₂CIV₁ particle images for refinement led to a map with a nominal resolution of 2.4 Å (Fig. 4a and Supplementary Fig. S2c–g). The map fits atomic models of CIII₂CIV₁ from previous studies with high fidelity[28–30,42,43].

The resolution of the current map is higher than that in previous studies of the *S. cerevisiae* supercomplex[28–30,42]. The higher resolution allowed us to resolve water molecules in two proton pathways of CIV[14,44–46]. Densities originating from nine water molecules are seen in the D proton pathway in subunit Cox1 of CIV, which connects the highly conserved Asp92 on the *n* side with another conserved protonatable residue, Glu243 (Supplementary Fig. S3), from where protons are transferred either to the catalytic site after binding of O₂ or are pumped toward the *p*-side of the membrane. In addition, two water molecules were identified in the K proton pathway, which connects Glu82, near the *n*-side surface of subunit Cox2, via the conserved Lys319 (Cox1), with the catalytic site (Supplementary Fig. S3). This pathway is used for proton transfer upon reduction of the oxidized catalytic site. The positions of all these water molecules are largely the same as those identified previously in high-resolution X-ray crystal

**Fig. 2 | Activity as a function of cyt. *c* concentration. a** QH₂:O₂ oxidoreductase activity of the *S. cerevisiae* CIII₂CIV₂ supercomplex at different cyt. *c* concentrations at 20 mM KCl buffer ($\lambda_D = 2.2$ nm). Data from measurements at 150 mM KCl ($\lambda_D = 0.8$ nm, blue)[30] are included for comparison. The data are normalized at the maximum activity at 50 μM cyt. *c*, which was ~50 electrons/s. Data are the average of three technical replicates measured with each of three different supercomplex preparations. Error bars are standard errors. **b** Hill equation fit (solid black line) of the 20 mM KCl activity data in the range $0.03 - 1\ \mu M$ cyt. *c*. The Hill coefficient was $n = 5 \pm 3$ ($K_A = 180$ nM), indicating the involvement of two or more cyt. *c* molecules in electron transfer between each CIII in CIII₂ and CIV. For comparison, the data at 150 mM KCl[30] were fitted with a Hill equation with $n = 1$ (solid blue line, $K_A = 1.7\ \mu M$).

concentration of cyt. *c* but 150 mM KCl[30]. At cyt. *c* concentrations above ~ 0.15 μM, the rate increased sharply to reach saturation at ~ 50 electrons/s at the highest cyt. *c* concentration (~ 50 μM), consistent with rate limiting cyt. *c*-mediated electron transfer between CIII₂ and CIV[30,40].

The data in the range 30 nM $-1\ \mu M$ cyt. *c* were fit with a Hill equation (Fig. 2b):

$$\theta = \frac{1}{1 + \left(\frac{K_A}{[\text{cyt}.c]}\right)^n} \tag{1}$$

where $\theta$ is the fraction supercomplex with an electronic link between each CIII of CIII₂ and CIV, and $K_A$ is the cyt. *c* concentration at which half the maximum rate is obtained, [cyt. *c*] is the concentration of added cyt. *c* and $n$ is the Hill coefficient. The solid line in Fig. 2b was obtained with $n = 5 \pm 3$ ($K_A = 180$ nM), i.e., a maximum rate of electron transfer requires binding of two or more cyt. *c* molecules. For comparison,

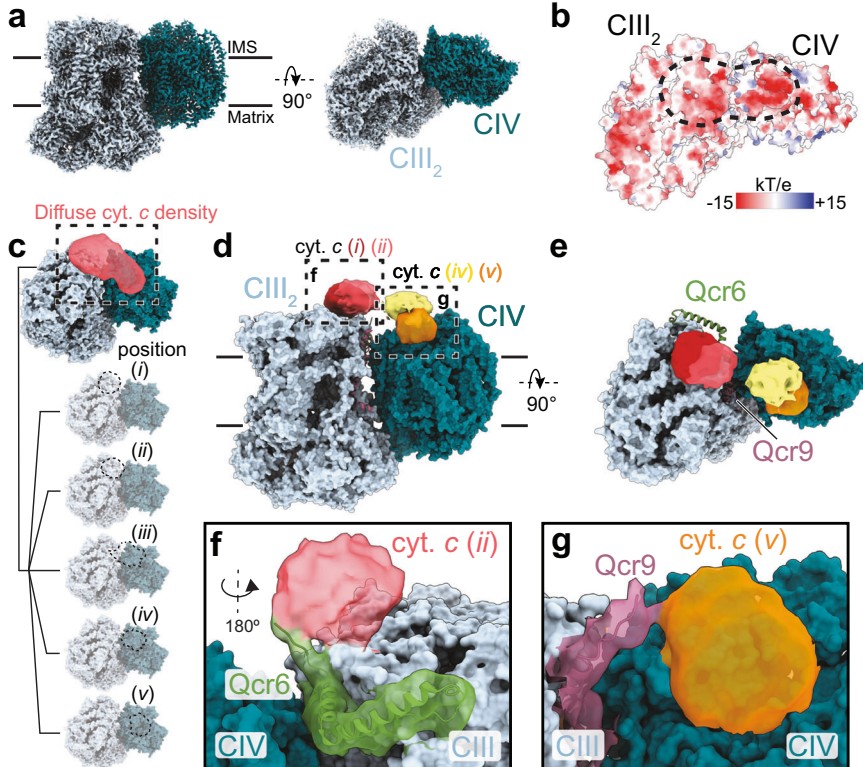

**Fig. 4 | Cryo-EM analysis. a** Cryo-EM map of the $CIII_2CIV_1$ supercomplex with the approximate position of the membrane indicated by black lines. IMS is intermembrane space. **b** Coulomb electrostatic surface representation of a supercomplex with the cyt. $c$-binding positions indicated by a dashed line. **c** $CIII_2CIV_1$ supercomplex with the cyt. $c$ density is shown in red. **c**–**e** Principal component analysis (PCA) or 3D classification revealed two possible cyt. $c$ positions at each CIII in $CIII_2$ ((*i*), (*ii*), dark and light red, respectively, in panels **d**, **e**) and CIV ((*iv*), (*v*), yellow and orange, respectively, in panels **d**, **e**). Panel (**c**), (*iii*) represents a class identified in between each CIII (in $CIII_2$) and CIV at 150 mM KCl[30]. Panels (**f**) and (**g**) are expanded views of the two dashed squares in panel (**d**) showing the positions of subunits Qcr6 with cyt. $c$ at $CIII_2$ and Qcr9 with cyt. $c$ at CIV, respectively.

structures of bacterial and mammalian CIV, but to the best of our knowledge, have not been resolved previously in the *S. cerevisiae* CIV.

As seen in the earlier presented $CIII_2CIV_{1/2}$ supercomplex structures[28,29,42,43], a number of lipid molecules, largely overlapping in space with those identified earlier, were resolved in the current structure (Supplementary Fig. S4a). However, in contrast to the earlier *S. cerevisiae* supercomplex models, we identified a cardiolipin (CL) at a site previously modeled as two phosphatidylethanolamine (PE) lipid molecules (Supplementary Fig. S4b, c), which is similar to the CL identified in *Yarrowia lipolytica* $CIII_2$[47]. In addition, we identified a CL outside of a V-shaped cleft formed by subunit Cox3 of CIV (Supplementary Fig. S4d, e), used for $O_2$ diffusion to the CIV catalytic site[48]. This CL molecule presumably interacts electrostatically with Arg67 (Cox3) and Arg25 (Cox7)[49], which may explain why it is seen at the lower ionic strength used in the current study.

As in our previous study performed at 150 mM KCl[30], we detected an elongated density corresponding to cyt. $c$ bound at different locations on the negatively charged $p$-side surface of the supercomplex, spanning between cyt. $c_1$ on each CIII in $CIII_2$ and $Cu_A$ on CIV. Using 3D classification, we next separated particle images into three distinct classes (Supplementary Fig. S5). These classes all showed density for cyt. $c$ on both the CIII and CIV portion of $CIII_2CIV_1$ at multiple binding positions (Fig. 4c–e, (*i*) and (*ii*), dark and light red, respectively) and CIV (Fig. 4c–e, (*iv*) and (*v*), yellow and orange, respectively). Two of these classes show density for multiple cyt. $c$ molecules simultaneously, but with one cyt. $c$ molecule visible at a much higher threshold ($\sigma = 4$) than the second cyt. $c$ molecule, which appears at a lower threshold ($\sigma = 2$) (Supplementary Fig. S5b, c). This difference in the density at these binding sites may be explained by a fraction of particle images within a class having only one bound cyt. $c$ molecule, or

that in each class, the position of the second cyt. $c$ is not static, which would yield a more diffuse density. The third class shows diffuse density in-between the $CIII_2$ and CIV parts of the $CIII_2CIV_1$ supercomplex (Supplementary Fig. S5d). Importantly, further classification of these three populations did not yield subpopulations with only a single cyt. $c$ bound, while in our previous dataset at 150 mM KCl, this separation was possible[30]. This observation is consistent with the kinetic data above, suggesting that at 20 mM KCl, there are $CIII_2CIV_1$ supercomplexes with multiple cyt $c$'s bound. Furthermore, the lack of a clear density at the intermediate position (Fig. 4c, (*iii*)) at 20 mM ionic strength is consistent with a higher affinity for cyt. $c$ binding to each CIII in $CIII_2$ and CIV, resulting in a higher probability for simultaneous binding of two cyt. $c$ molecules, thereby lowering the probability of binding of cyt. $c$ at the intermediate position where it would clash with cyt. $c$ molecules bound at either one CIII in $CIII_2$ or CIV. Binding of cyt. $c$ at two positions at each CIV and at each CIII in $CIII_2$ at low ionic strength was observed previously and suggested to be involved in electron transfer by cyt. $c$ 2D diffusion also within the mammalian[50] and plant[51] supercomplexes. In summary, the present study, together with the earlier study performed at 150 mM KCl[30], reveals five different cyt. $c$ positions at the negatively charged $p$-side surface of the supercomplex (Fig. 4c). Positions (*i*) (Fig. 4d, e, dark red) and (*ii*) (Fig. 4d, e, light red) at each of the CIII monomers in $CIII_2$ are observed at 20 mM KCl. Position (*i*) was also observed at 150 mM KCl[30] and is located closer to cyt. $c_1$ than position (*ii*), which is observed exclusively at low ionic strength. The intermediate cyt. $c$ position (*iii*) between each CIII monomer in $CIII_2$ and CIV (Fig. 4c) was observed clearly only at 150 mM KCl in our earlier study[30], while at low ionic strength, it is seen as a diffuse density (Supplementary Fig. S5d). Positions (*iv*) (Fig. 4de, yellow) and (*v*) (Fig. d, e, orange) at CIV are observed at 20 mM KCl.

Position ($v$), closer to $Cu_A$ than position ($iv$)(Fig. 4d), was also observed at 150 mM KCl. Position ($v$) overlaps with the position seen in the crystal structure of the CIV-cyt. $c$[52] co-complex. The cyt. $c$ position observed in the yeast CIII$_2$-cyt. $c$ co-complex[53] is slightly shifted away from CIV compared to position ($i$) in the current study.

The 3D classification made it possible to resolve density for an additional 3–4 of the 73 residues at the N terminus of Qcr6, which, to the best of our knowledge, could not be resolved previously in either X-ray crystal structures of CIII$_2$[54] or cryo-EM structures of the CIII$_2$CIV$_{1/2}$ supercomplex[28–30,42,43]. This N-terminal sequence of Qcr6 is referred to as the "hinge peptide"[55], which is predicted to be unstructured (Supplementary Fig. S6a, c). In the current study, the additional 3–4 residues are in close contact with cyt. $c$ when it is near CIII$_2$ (Fig. 4d–f). These residues could not be resolved previously, presumably because without bound cyt. $c$ the entire Qcr6 loop can adopt an ensemble of positions. The loop is composed of a majority negatively charged residues where residues 41–73 are to ~ 90 % Glu or Asp (Supplementary Fig. S6a), which have been shown to be essential for binding of the positively charged cyt. $c$ to CIII$_2$[54–57]. Furthermore, a recent combined cryo-EM and theoretical study suggested that the disordered loop of the Qcr6 subunit facilitates substrate diffusion between each CIII monomer of CIII$_2$ and the adjacent CIV[58]. The cryo-EM data from the current study, where the Qcr6 density interacts with the bound cyt. $c$ (Fig. 4d, f), is consistent with the earlier observations.

As seen in Fig. 4d–g, we also observed density that is attributed to 7-8-amino-acid residues C-terminal segment of subunit Qcr9, which interacts with cyt. $c$ bound at CIV. A weak density, presumably originating from this Qcr9 loop, is also seen in another particle class for cyt. $c$ at CIII$_2$, but this density could not be modeled confidently. The C-terminal sequence from Qcr9 is also composed of a majority (~ 80%) negatively charged residues (Supplementary Fig. S6b), which would explain why it could not be observed in the absence of cyt. $c$.

## Discussion

Diffusion of cyt. $c$ between each CIII monomer in CIII$_2$ and the neighboring CIV within the CIII$_2$CIV$_{1/2}$ supercomplex is supported by electrostatic interactions between the positively charged docking surface of cyt. $c$ and the negatively charged supercomplex surface. Such interactions are manifested by an association of cyt. $c$ with supercomplexes upon isolation from e.g., *Ustilago maydis*[59] or *S. cerevisiae*[20,30–34,60]. Binding of cyt. $c$ must be sufficiently strong to maintain proximity between the ligand and the supercomplex, yet weak enough to allow cyt. $c$ sliding along the surface. Electron transfer from QH$_2$ to O$_2$ within the supercomplex is rate limited by the cyt. $c$-mediated electron transfer between CIII$_2$ and CIV[30,40]. Hence, interactions between cyt. $c$ and the supercomplex surface govern the quinol oxidation:O$_2$ reduction activity of the supercomplex.

The distinct sharp increase in the supercomplex turnover rate at 0.1–0.4 μM cyt. $c$ (cyt. $c$:CIV ratio of 5–20) at a salinity of 20 mM contrasts the behavior observed previously at a salinity of 150 mM[30] (Fig. 2a). A fit of the steady-state kinetic data with a Hill equation in the transition range (Fig. 2b), shows that at least two cyt. $c$ molecules bind cooperatively and are involved in electron transfer. Two scenarios are proposed to explain the sharp transition in the 0.1–0.4 μM cyt. $c$ range: ($i$) Binding of a second cyt. $c$ molecule results in shorter 2D diffusion distances. ($ii$) Initially, two cyt. $c$ molecules are recruited, and in the sharp transition, a third molecule binds between the two pre-bound cyt. $c$ molecules to connect the electron donor and acceptor sites electronically. With a diameter of ~ 35 Å of each cyt. $c$, three cyt. $c$ molecules are sufficient to provide an electron transfer wire connecting cyt. $c_1$ and $Cu_A$. Thus, according to scenario ($ii$), the binding of a third cyt. $c$ molecule would yield the maximum possible electron-transfer rate. However, the supercomplex activity is not saturated after the sharp transition in the 0.1–0.4 μM cyt. $c$ range, but increases further to reach a maximum rate at higher cyt. $c$ concentrations (Fig. 2a).

Hence, we propose that the transition in the range 0.1 μM – 0.3 μM cyt. $c$ involves two bound cyt. $c$ molecules (scenario ($i$)), one at a CIII monomer in CIII$_2$ and a second cyt. $c$ at the adjacent CIV. Electron transfer between the CIII monomer in CIII$_2$ and CIV would take place by 2D diffusion of the two cyt. $c$ molecules such that they would exchange electrons upon interaction at the surface. The further increase at >1 μM cyt. $c$ involves binding of a third cyt. $c$ molecule (scenario ($ii$)). We note that the maximum turnover rate (Fig. 2a) is obtained at concentrations similar to native conditions in *S. cerevisiae*, i.e., ~ 100 μM[61].

Involvement of two cyt. $c$ molecules is also supported by the cryo-EM data, which suggest that at low salinity and a cyt. $c$:CIV ratio of 12, i.e., within the sharp transition in Fig. 2a, a subpopulation of the particles harbor two cyt. $c$ molecules per CIII monomer-CIV pair (Fig. 4). Furthermore, the cryo-EM data indicate that the intermediate cyt. $c$ position between each CIII monomer in CIII$_2$ and the adjacent CIV (($iii$), in Fig. 4c), clearly seen at 150 mM KCl in our earlier study[30], is less populated at 20 mM KCl (**see** Supplementary Fig. S5). This observation is consistent with a scenario where two cyt. $c$s are bound, one at each of CIV and a CIII monomer in CIII$_2$, which by steric interference lowers the probability for binding of a cyt. $c$ at the intermediate position.

Combined cryo-EM and kinetic data from our earlier study[30] indicate that at 150 mM KCl, electron transfer from each CIII in CIII$_2$ to CIV occurs by sliding one cyt. $c$ molecule along the negatively charged supercomplex surface. This mechanism yields an electron-transfer rate between each CIII in CIII$_2$ and CIV of ~ 20 s$^{-1}$ at 150 mM KCl and a cyt. $c$:CIV ratio of 5. The data from the current study show that the supercomplex activity at cyt. $c$:CIV ratios < 5 (cyt. $c$ concentrations < 0.1 μM) are lower at 20 mM than at 150 mM KCl (see Fig. 2a). This difference is presumably due to a tighter electrostatic association of cyt. $c$ with the supercomplex surface at the lower ionic strength. At low cyt. $c$:CIV ratios electron transfer between each CIII in CIII$_2$ and CIV presumably takes place by 2D diffusion, which would be slowed by a tighter association of both reduced and oxidized cyt. $c$ with the supercomplex surface.

We also measured the concentration reduced cyt. $c$ in solution as a function of the cyt. $c$ concentration (Fig. 3). This concentration increased with increasing concentration of added cyt. $c$, but the fraction of reduced cyt. $c$ decreased (Fig. 3, inset). Both the concentration and fraction reduced cyt. $c$ were larger at 20 mM than at 150 mM salt concentration. A larger fraction or concentration of reduced cyt. $c$ at 20 mM than at 150 mM KCl could be explained by a lower activity of CIV at 20 mM than at 150 mM KCl. However, the CIV activity is presumably not rate limiting because the maximum O$_2$-reduction activity of CIV is larger than the QH$_2$:O$_2$ oxidoreductase activity of the supercomplex. Instead, we speculate that the fraction of cyt. $c$ that is reduced in solution is kept in this state by transient docking of an oxidized cyt. $c$ molecule that remove an electron from the supercomplex. Assuming electrostatic interactions also occur between cyt. $c$ molecules, the probability for a solution cyt. $c$ to interact with a supercomplex-bound cyt. $c$ is higher at a low ionic strength than at high ionic strength, which would explain the larger fraction reduced cyt. $c$ at the lower ionic strength. Furthermore, the probability for electron transfer via multiple cyt. $c$ molecules at the supercomplex surface increases with increasing cyt. $c$ concentration. As the probability for reduction of cyt. $c$ in solution would decrease under conditions of electron transfer via a surface-associated wire, the fraction reduced cyt. $c$ would decrease with increasing cyt. $c$ concentration. We also note that the data show that the steady-state concentration of reduced cyt. $c$ at 100 μM cyt. $c$ qualitatively agrees the 10 % reduced cyt. $c$ in *S. cerevisiae*[62,63].

In a recent study, Vercellino and Sazanov studied electron transfer between CIII$_2$ and CIV in a mammalian CIII$_2$-CIV supercomplex as well as in a mixture of isolated CIII$_2$ and CIV complexes at low ionic strength (20 mM Hepes buffer)[33]. As also observed by Moe et al.[30], they found that the quinol oxidation:O$_2$ reduction activity, mediated by cyt. $c$

electron transfer, was higher with the supercomplex than with a mixture of CIII$_2$ and CIV. Interestingly, as in the present study, Vercellino and Sazanov observed that at 5–10 nM cyt. $c$ (cyt. $c$: supercomplex ratio of 2.5–5), the supercomplex had "barely any activity" (c.f. ~10 % activity in the current study, Fig. 2), but at cyt. $c$ concentrations in the µM range the activity increased dramatically[33]. The authors suggested that the 2D diffusion mechanism involves two or more bound cyt. $c$ per supercomplex[33], but that this mechanism is applicable uniquely to mammalian mitochondria. Based on the findings in the present study, we suggest that the mechanism is the same in *S. cerevisiae* and in mammals, but the difference in rates at 5–10 nM and µM cyt. $c$, respectively, is explained by the low ionic strength used in ref. 33, which would yield slow 2D diffusion of a single cyt. $c$ bound to the supercomplex (at 5–10 nM cyt. $c$).

A re-analysis of our data obtained at 150 mM KCl[30] shows that in the intermediate position (Fig. 4c, position (*iii*)), density was observed for both Qcr6 and Qcr9 interacting with cyt. $c$. Interestingly, the Qcr6 density at this cyt. $c$ position is located on the opposite side of cyt. $c$ compared to the density observed from Qcr6 when cyt. $c$ is bound to CIII$_2$ at 20 mM KCl (Fig. 4f). The resolution of the cryo-EM data is too low to allow determining the rotational pose of cyt. $c$ itself when bound to CIII$_2$ or CIV, respectively. Assuming that Qcr6 binds at the same position on cyt. $c$ at 20 mM and 150 mM salt, the data suggest that cyt. $c$ would rotate while moving along the supercomplex surface from each CIII in CIII$_2$ to the adjacent CIV. To address this question, we compared the positions of cyt. $c$ at a CIII monomer in CIII$_2$ and at CIV based on the X-ray crystal structures of the CIII$_2$-cyt. $c$[53] and CIV-cyt. $c$[52] co-complexes from *S. cerevisiae* and bovine heart, respectively (Supplementary Fig. S7). When superimposed on the *S. cerevisiae* supercomplex, the data show that cyt. $c$ would rotate while moving between the two complexes, which is consistent with binding of Qcr6 and Qcr9 at different rotational states of cyt. $c$ in the current study (Supplementary Fig. S8). The interaction with Qcr6 and Qcr9 could facilitate sliding of a single cyt. $c$ along a well-defined trajectory between the cyt. $c$-binding sites at cyt. $bc_1$ and Cyt$c$O, respectively, as also discussed by ref. 58 (Supplementary Fig. S8). Alternatively, interactions between Qcr6/9 and cyt. $c$ would facilitate correct positioning of one cyt. $c$ molecule at each of CIII$_2$ and CIV that would fluctuate along well-defined trajectories to adopt the correct pose for electron exchange between two bound cyt. $c$ molecules to transfer an electron along the supercomplex surface.

As outlined in the Introduction section, cellular protein-protein interactions are commonly of electrostatic origin, nonspecific and only manifest themselves under densely crowded conditions[2,4–6]. In eukaryotes the average proteome is optimized to carry a net negative charge. In contrast, the dipolar cyt. $c$ carries a net positive charge[7], which slows diffusion within the densely-packed negatively charged proteome. Specific interactions between the positively charged cyt. $c$ and the highly negatively charged supercomplex surface would thus outcompete the unspecific interactions with non-partner proteins thereby promoting electron transfer within a CIII$_2$CIV$_{1/2}$ supercomplex by 2D diffusion. The combined cryo-EM and kinetic data show that the supercomplex system is highly responsive to the electrostatic environment and that lowering the ionic strength yields an electron-transfer mechanism that involves more than one cyt. $c$, which yields a system that can be tightly regulated by changes in the cyt. $c$:supercomplex ratio.

## Methods

### Strain and cell growth
*S. cerevisiae* strain BY 4741 with a FLAG tag on CIV subunit Cox6 was used[41]. Yeast cultures were grown in YPG (1% yeast extract, 2% peptone and 2% glycerol) at 30 °C, 180 rpm (Shaker Innova 44 R).

### Preparation of mitochondrial membranes
Cells were harvested at 6500 × $g$ (5 min, 4 °C) by centrifugation and washed with 50 mM KH$_2$PO$_4$ at pH 7. The pellet was resuspended in a buffer containing 650 mM mannitol, 50 mM KH$_2$PO$_4$ and 5 mM EDTA at pH 7.4. Cells were lysed by mechanical disruption (Constant Systems cell disrupter) at a pressure of 35 kpsi and cell debris was removed by centrifugation at 5600 × $g$ (20 min, 4 °C). The supernatant was then centrifuged at 120,000 × $g$ for 1 h at 4 °C. The pellet was resuspended and homogenized in 100 mM KCl, 50 mM KH$_2$PO$_4$ and 5 mM EDTA at pH 7.4. It was then centrifuged at 120,000 × $g$ for 30 min at 4 °C, washed, and centrifuged twice (see above) in 50 mM KH$_2$PO$_4$ and 5 mM EDTA at pH 7.4. Finally, the pellet was homogenized in 50 mM KH$_2$PO$_4$ at pH 7.4, frozen in liquid nitrogen and stored at −80 °C until use.

### Isolation of supercomplexes
Membrane fragments were diluted to a total protein concentration of 2 mg/mL in 100 mM KCl, 50 mM KH$_2$PO$_4$ at pH 7.4, and 0.5 % (weight/volume) GDN (GDN101; Anatrace) and solubilized overnight at 4 °C. The membrane lysate was centrifuged at 140,000 × $g$ for 90 min at 4 °C. The supernatant was diluted to reach a final GDN concentration of 0.2 % and the sample was concentrated by centrifugation to a volume of ~70 mL using a 100-kDa molecular weight cut-off concentrator (Merck Millipore). The concentrated extract was incubated with 1 mL anti-FLAG M2 resin (Sigma-Aldrich) for 2 h at 4 °C. The anti-FLAG M2 resin with bound protein was washed with 30 mL of buffer (150 mM KCl, 20 mM KH$_2$PO$_4$ at pH 7.4 and 0.01 % GDN) and eluted with 0.1 mg/mL FLAG peptide (Sigma-Aldrich), 150 mM KCl, 20 mM KH$_2$PO$_4$ at pH 7.4, and 0.01% GDN. Eluted supercomplex was concentrated with a 100-kDa molecular weight cutoff concentrator (Merck Millipore) and further purified by size exclusion chromatography with an Äkta Pure M25 (GE Healthcare) operated at 4 °C while monitoring absorbance changes at 280 nm and 415 nm. The sample was loaded on a Superose 6 Increase 10/300 GL column (GE Healthcare) equilibrated with 150 mM KCl, 20 mM KH$_2$PO$_4$ at pH 7.4, and 0.01 % GDN. Fractions of 250 µL were collected, and those containing both CIII$_2$ and CIV were pooled and concentrated as described above.

### UV-Visible difference spectroscopy
Optical absorption spectra were recorded with a Cary 100 UV-VIS spectrophotometer (Agilent Technologies) to calculate the concentration of CIII$_2$ and CIV in the sample. A reduced minus oxidized difference spectrum was obtained by subtracting the "as isolated" (oxidized) spectrum from that obtained after addition of a small aliquot of sodium dithionite as a reducing agent. Peaks were fitted for the different heme groups, heme $c_1$ (554 nm), hemes $b_H$ and $b_L$ (562 nm), and hemes $a$ and $a_3$ (603 nm), and their concentrations were calculated using the following difference absorption coefficients: $\Delta\varepsilon_{554} = 21$ mM$^{-1}$cm$^{-1}$, $\Delta\varepsilon_{562} = 51.2$ mM$^{-1}$cm$^{-1}$, and $\Delta\varepsilon_{603} = 25$ mM$^{-1}$cm$^{-1}$[64,65].

### Activity measurements
Decylubiquinone (DQ, Sigma-Aldrich) was dissolved in 99.8% ethanol to reach a concentration of 20 mM. Crystals of sodium borohydride (NaBH$_4$; Sigma-Aldrich) were added and the sample was incubated on ice until it turned colorless, which is indicative of formation of the reduced DQH$_2$. When the solution was clear, 10 − 20 µL of 5 M HCl (depending on the amount of NaBH$_4$ used for reduction) was added until the pH was ~3. The sample was then centrifuged for 10 min at 10,000 × $g$ to remove any excess undissolved NaBH$_4$ crystals, and the supernatant containing DQH$_2$ was collected.

The activity of CIII$_2$ was measured by following in time absorbance changes at 550 nm upon addition of DQH$_2$ and KCN to block the oxidation of cyt. $c$ by CIV. First, the baseline absorbance was recorded using 50 µM oxidized cyt. $c$ from *S. cerevisiae* (Sigma-Aldrich), CIII$_2$CIV$_{1/2}$ supercomplex (equivalent to 10 nM CIII$_2$CIV$_2$) and 2 mM KCN in 20 mM KCl, 20 mM KH$_2$PO$_4$ pH 7.4 and 0.01% GDN. The reaction was initiated by adding 100 µM DQH$_2$.

CIV activity was measured using a Clark-type oxygen electrode (Oxygraph; Hansatech) operated at 25 °C to monitor the oxygen-

reduction rate. A baseline was first recorded using 50 μM oxidized cyt. *c* from *S. cerevisiae*, 10 mM ascorbate, 100 μM N, N, N', N'-tetramethyl-p-phenylenediamine (TMPD) in 20 mM KCl, 20 mM $KH_2PO_4$ pH 7.4 and 0.01% GDN. The reaction was initiated by adding $CIII_2CIV_{1/2}$ supercomplex at a concentration equivalent to 10 nM $CIII_2CIV_2$. The solution volume of the oxygraph chamber is 1 ml.

The activity of *S. cerevisiae* $CIII_2CIV_{1/2}$ supercomplex was measured by monitoring the oxygen-reduction rate with a Clark-type oxygen electrode (Oxygraph; Hansatech) operated at 25 °C. Measurements were done at a concentration equivalent to 10 nM $CIII_2CIV_2$ and varying the concentration of *S. cerevisiae* cyt. *c*, in 20 mM KCl, 20 mM $KH_2PO_4$ pH 7.4, and 0.01 % GDN. The reaction was initiated by adding 100 μM $DQH_2$. A control measurement was done by monitoring the $O_2$ reduction rate upon addition of all components, except cyt. *c* to the oxygraph chamber, and reduction rate was negligible. To determine the steady state concentration of reduced cyt. *c*, we monitored absorbance at 550 nm. It was normalized to that of a reduced (using dithionite) - oxidized difference absorbance at 550 nm.

### Grid preparation and cryo-electron microscopy
Purified supercomplex was exchanged into 20 mM KCl, 20 mM $KH_2PO_4$ pH 7.4, and 0.01 % GDN with a 100-kDa molecular weight cutoff concentrator (Merck Millipore). A sample consisting of 3 μL of 10.5 μM supercomplex, supplemented with 126 μM *S. cerevisiae* cyt. *c* (Sigma-Aldrich) was applied to glow-discharged (20 mA, 30 s using PELCO easiGlow) Quantifoil R2/2 Cu grids manually covered with a continuous film of ~ 5 nm carbon and vitrified by plunge-freezing into liquid ethane after blotting for 9 s at 4 °C and 100% humidity using a Vitrobot Mark IV (Thermo Fisher Scientific). Cryo-EM data were collected using a Titan Krios G3 electron microscope (Thermo Fisher Scientific) operated at 300 kV, equipped with a Gatan BioQuantum energy filter and a K3 Summit direct electron detector (AMETEK). Automated data collection was done with the EPU software package (Thermo Fisher Scientific). A dataset of 13,453 movies, each consisting of 35 exposure fractions was collected at a nominal magnification of 105,000x, corresponding to a calibrated pixel size of 0.85 Å. The camera exposure rate and the total exposure of the specimen were 17.6 $e^-$/pixel/s and 40 $e^-/Å^2$, respectively.

### Cryo-EM image processing
All image analysis was performed with cryoSPARC v3[66]. Exposure fractions were aligned with MotionCor2[67] and contrast transfer function (CTF) parameters were estimated in patches. Particles were picked by Blob Picker and picked again by the template generated from Blob-picked particles. Dataset was cleaned by 2D classification and 3D classification, taking only classes corresponding to both $CIII_2CIV_1$ and $CIII_2CIV_2$ after each round. $CIII_2CIV_1$ and $CIII_2CIV_2$ were combined to 196,457 particles and used to refine a map of $CIII_2CIV_1$ with nonuniform refinement to 2.4 Å[68]. The particle images were subjected to 3D classification using a mask over the region corresponding to cyt. *c*. 3D variability analysis (3DVA)[69] with the same mask yielded similar results (3D classification was used in Fig. 4). Masked local refinement of $CIII_2$ enforcing $C_2$ symmetry yielded a map at 2.3 Å resolution and masked local refinement of CIV yielded a map at 2.5 Å resolution. Symmetry expansion was used on the $CIII_2$ map before analyzing the region of the map corresponding to cyt. *c* using a mask on the particular region and subjected to either 3DVA and 3D classification. Both analyses showed similar output and the 3DVA was used in Fig. 4).

### Model building and refinement
A composite map was created for model building using UCSF Chimera[70]. Locally refined maps for $CIII_2$ and CIV were aligned with the full supercomplex map and combined to create the composite map. The previously determined yeast $CIII_2IV_2$ supercomplex structure (PDB

ID code 6HU9)[28] was fitted into the composite map using Chimera[70]. This model was further refined with Coot[71] and real-space refinement in Phenix[72].

### Reporting summary
Further information on research design is available in the Nature Portfolio Reporting Summary linked to this article.

## Data availability
The cryo-EM maps and atomic coordinates have been deposited in the Electron Microscopy Data Bank and Protein Data Bank, under accession numbers EMD-19963, EMD-19964, EMD-19965, EMD-19966 and PDB 9ETZ, respectively. Source data are provided with this paper.

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

## Acknowledgements

This work was supported by the Knut and Wallenberg Foundation (M.O., P.B.), the Swedish Research Council (MO, PB), and the Canadian Institutes of Health Research Project PJT191893 (J.L.R.). The data were collected at the Cryo-EM Swedish National Facility funded by the Knut and Alice Wallenberg, Family Erling Persson and Kempe Foundations, SciLifeLab, Stockholm University and Umeå University. J.D.T. was supported by a postdoctoral fellowship from the Canadian Institutes of Health Research and J.L.R. was supported by the Canada Research Chairs program.

## Author contributions

M.O., P.B., and A.M. conceived the project. A.P.L. prepared the supercomplex, performed the kinetic experiments, and interpreted the data, guided by A.M., M.O., and P.B. F.W. prepared cryo-EM samples performed electron microscopy and initial image analysis. A.M. and J.D.T. performed image analysis and interpreted the cryo-EM data. A.M. prepared figures. J.L.R advised on image analysis and interpretation of the structures. A.M. and P.B. wrote the manuscript with input from the other authors.

## Funding

## Competing interests

The authors declare no competing interests.
