## [Peer Review File · Nature Communications]

Reviewers' Comments:

Reviewer #1:

Remarks to the Author:

The authors for the first time performed steady-state kinetic studies and Cryo-EM structural analysis to investigate electrostatic interactions between cytochrome c and the supercomplex composed of CIII2 and CIV respiratory complexes from *Saccharomyces cerevisiae* at low salinity, 20 mM monovalent salt, as the recent studies (“Colloidal stability of the living cell”, by Wennerström et al 2020) demonstrated that this concentration rather than 150 mM used previously, reflects the physiological condition inside the living cells containing large polyanionic structures. This work is the next step in the study of the respiratory supercomplexes. The study is very well designed and the results obtained by the authors are remarkable. Kinetic studies demonstrate that maximum rate of electron transfer requires binding of two or more cyt. c molecules (show a sharp transition with a Hill coefficient ≥ 2) and structural studies show multiple cyt. c molecules bound along the supercomplex surface. The cryo-EM structure of the *S. cerevisiae* CIII2CIV1/2 supercomplex with bound multiple molecules of cyt c was obtained at 2.4 Å resolution. This is the highest resolution structure comparing with previously reported similar structures (6HU9 3.35 Å, 6GIQ 3.32 Å, 8E7S 3.2 Å, 8EC0 3.3 Å, 6YMX 3.17 Å, 6T15 3.29 Å, 6T0B 2.8 Å). The authors were able to get such a high-resolution structure of the supercomplex because under condition of low salinity previously unresolved negatively charged highly flexible loop of Complex III subunits Qcr6 and Qcr9 became structured upon interaction with multiple cytochrome c molecules. Thus, kinetic and structural data together allowed the authors to create a new model for electron transfer from CIII2 to CIV by cytochromes c in the supercomplex via “bridging”. In addition, the authors were able to identify water molecules in proton pathways of CIV as well as previously unresolved cardiolipin molecules.

The presented work has strong conclusions based on using high experimental techniques and data analysis. The obtained results are important for the advance in the field of mitochondria energy production in living organisms. I strongly recommend this work for publication in Nature Communications.

I have only few minor comments:

In introduction:

“For example, in the yeast *S. cerevisiae* these supercomplexes are composed of a CIII2 dimer flanked by one or two copies of CIV (CIII2CIV1/2)”

Complex III abbreviation is CIII2

In the text of the manuscript the authors use CIII2 and also CIII; and just III. For readers not from the field it might create the impression that these are different structures. Also: III-IV pair,

CIII2CIV1/2 supercomplex and III2IV1/2 supercomplex

Another example – the legend to the Supplementary Figure S5. Cryo-EM 3D classification

The authors use: CIII2CIV1 , also both CIII and CIII2

I suggest to include into discussion the schematic presentation of the electron transfer from Complex III to Complex IV in the supercomplex through the multiple cytochromes c associated with bridge between these complexes (built by the domains of Qcr6 and Qcr9 subunits).

Reviewer #2:

Remarks to the Author:

The manuscript “Electron transfer in the respiratory chain at low salinity” describes a functional and cryo-EM study of yeast respiratory supercomplex under lower than usual ionic strength (20 mM vs 150 mM). It is argued that these low salt conditions are closer to physiological, which seems to make sense overall. Unlike previously, electron transfer rates show a sharp transition with a Hill coefficient ≥ 2 . Consistent with this cooperativity, cryo-EM studies show two molecules of cyt c bound per supercomplex, and two protein loops become resolved upon this binding. Overall, this is a well-written and illustrated study, improving our knowledge on the details of cyt c interactions with the supercomplex. However, this seems to be a rather incremental advance, not a major breakthrough in the area.

Reviewer #3:

Remarks to the Author:

The manuscript by Lobez et al. is an interesting work that brings noteworthy results on the functional role of the mitochondrial respiratory supercomplexes by delving into the diffusion mechanism of cytochrome c (cyt. c) between Complex_III and Complex_IV within the yeast III+IV supercomplex of *S. cerevisiae*.

Notably, the authors provide combined cryo-EM and kinetic data supporting the conclusion that the III+IV supercomplex system is highly responsive to the electrostatic environment and that lowering the ionic strength can reveal a substrate channeling mechanism involving more than one tightly bound cyt. c molecule.

The topic is of great interest for the scientific community who is still largely debating the kinetic advantage of substrate channeling in supercomplexes. Indeed, it is ascertained in the literature that profound differences exist in different organisms, so it would be

even more interesting if the authors could further discuss their results in consideration also of recent publications about cyt. c channeling in mammalian mitochondrial respirasomes.

As minor revisions before publication, we recommend:

- expressing all cyt. c concentrations as " μM " values throughout the text and in the figures, in order to make the comparison between values more immediate;
- changing a typo in the "Isolation and activity of the supercomplex" section, last line: specify "150 mM KCl" in "...fit with a Hill equation to the data at 150 mM" if deemed appropriate.

Reviewer #4:

Remarks to the Author:

In the paper entitled "Electron transfer in the respiratory chain at low salinity," the authors explored the role of salinity in the electrostatic interaction between CIII-CIV supercomplexes and cytochrome c during electron flux. The work is exciting and well-conducted. This reviewer has some questions.

1) The authors recover the idea that the cellular concentration of chlorine is low compared to that of potassium (Wennerström, H., et al. Proc. Natl. Acad. Sci. USA (2020): 117, 10113-10121); however, the total amount of positive and negative charge is equal in intracellular fluid, in line with the principle of electro-neutrality. In this sense, the amount of Na (8 mEq/L) + K (148 mEq/L) + Mg (35 mEq/L) + Others (7 mEq/L) = Cl (17 mEq/L) + HCO₃ (9.3 mEq/L) + Protein (35 mEq/L) + Phosphate and Organic Acid (137 mEq/L). In essence, all charges (positive or negative) have their countercharge.

In the text, the authors assume that the cytoplasmic amount of free anions (i.e. Cl) is lower than free cations (i.e. K). In the cytoplasm, all negative charges (protein included) are spontaneously associated with positive charges, and the free ions concentrations must be very low. Please explain why assume that free positive charges are higher than negative charges in the cytoplasm without energy expenditure.

2) Authors showed that the activity of complex III is similar in the presence of 20 mM KCl + 50 μM cyt c (80 ± 6 e/s) or 150 mM KCl + 50 μM cyt c (90 ± 20 e/s); while the activity of complex IV from supercomplexes rise when KCl increased in the presence of ascorbate plus TMPD: 20 mM KCl and cyt c (120 ± 10 e/s) and 150 mM KCl (450 ± 20 e/s). The data showed that the rate-limiting step in the electron flux is complex III. Similar to the metabolic flux analysis showed.

Then, in Fig. 2 the authors suggested that the reduction of salinity increased the affinity of the supercomplex III/IV for the cytochrome c (from 1.7 μM to 180 nM). Complex III increases its affinity for cyt c oxidized, or complex IV increases its affinity for cyt c reduced? Please explain.

In this sense, kinetic analysis indicates that at low salinity the $n = 5$, suggests kinetic cooperativity or, as the authors said, the electron flux involves two or more cyt c molecules. If cyt c reduced is retained next to the active site, increasing enzyme affinity is not involved. Please explain.

The equation presented by the authors in the manuscript can be modified as the equation for an essential enzyme activator. Can the reduced cyt c act as an activator of the supercomplex?

In Fig. S1 the authors showed the QH₂ oxidation-O₂ reduction rate over time (140 nM/s, as calculated from the oxygen consumption record). Please include the oximeter chamber volume to determine the quinol:oxygen stoichiometry (mol/mol) and compare the activity of the supercomplex and the activity of individual complexes. Do these activities agree with the electron flux?

3) In Fig. 3, authors showed that cyt c reduced increase vs cyt c addition, in the presence of 20 mM KCl. Simultaneously, CIV activity was decreased with 20 mM KCl, in contrast to its high activity with 150 mM KCl. The accumulation of cyt c reduced could be the effect of low salinity on the CIV? Please explain.

Finally, Isolated supercomplexes showed QH₂:O oxidoreductase activity without cyt c addition? If did, this means that cyt c is tightly bound to the supercomplex, as reported previously by Reyes-Galindo et al. (B.B.A-Bioenergetics. 2019; 1860: 618–627) and can be a reinforcement of the hypothesis.

Reviewer #1

The authors for the first time performed steady-state kinetic studies and Cryo-EM structural analysis to investigate electrostatic interactions between cytochrome c and the supercomplex composed of CIII2 and CIV respiratory complexes from *Saccharomyces cerevisiae* at low salinity, 20 mM monovalent salt, as the recent studies (“Colloidal stability of the living cell”, by Wennerström et al 2020) demonstrated that this concentration rather than 150 mM used previously, reflects the physiological condition inside the living cells containing large polyanionic structures. This work is the next step in the study of the respiratory supercomplexes. The study is very well designed and the results obtained by the authors are remarkable. Kinetic studies demonstrate that maximum rate of electron transfer requires binding of two or more cyt. c molecules (show a sharp transition with a Hill coefficient ≥ 2) and structural studies show multiple cyt. c molecules bound along the supercomplex surface. The cryo-EM structure of the *S. cerevisiae* CIII2CIV1/2 supercomplex with bound multiple molecules of cyt c was obtained at 2.4 Å resolution. This is the highest resolution structure comparing with previously reported similar structures (6HU9 3.35 Å, 6GIQ 3.32 Å, 8E7S 3.2 Å, 8ECO 3.3 Å, 6YMX 3.17 Å, 6T15 3.29 Å, 6TOB 2.8 Å). The authors were able to get such a high-resolution structure of the supercomplex because under condition of low salinity previously unresolved negatively charged highly flexible loop of Complex III subunits Qcr6 and Qcr9 became structured upon interaction with multiple cytochrome c molecules. Thus, kinetic and structural data together allowed the authors to create a new model for electron transfer from CIII2 to CIV by cytochromes c in the supercomplex via “bridging”. In addition, the authors were able to identify water molecules in proton pathways of CIV as well as previously unresolved cardiolipin molecules. The presented work has strong conclusions based on using high experimental techniques and data analysis. The obtained results are important for the advance in the field of mitochondria energy production in living organisms. I strongly recommend this work for publication in Nature Communications.

I have only few minor comments:

In introduction: “For example, in the yeast *S. cerevisiae* these supercomplexes are composed of a CIII2 dimer flanked by one or two copies of CIV (CIII2CIV1/2)”. Complex III abbreviation is CIII2. In the text of the manuscript the authors use CIII2 and also CIII; and just III. For readers not from the field it might create the impression that these are different structures. Also: III-IV pair, CIII2CIV1/2 supercomplex and III2IV1/2 supercomplex

We have modified the text and refer in the revised version only to "CIII₂" or "monomer of CIII₂".

Another example – the legend to the Supplementary Figure S5. Cryo-EM 3D classification The authors use: CIII2CIV1, also both CIII and CIII2.

We have modified the legend to address this point.

I suggest to include into discussion the schematic presentation of the electron transfer from Complex III to Complex IV in the supercomplex through the multiple cytochromes c associated with bridge between these complexes (built by the domains of Qcr6 and Qcr9 subunits).

We have added an additional supplementary Figure S8.

Reviewer #2

The manuscript "Electron transfer in the respiratory chain at low salinity" describes a functional and cryo-EM study of yeast respiratory supercomplex under lower than usual ionic strength (20 mM vs 150 mM). It is argued that these low salt conditions are closer to physiological, which seems to make sense overall. Unlike previously, electron transfer rates show a sharp transition with a Hill coefficient ≥ 2 . Consistent with this cooperativity, cryo-EM studies show two molecules of cyt c bound per supercomplex, and two protein loops become resolved upon this binding. Overall, this is a well-written and illustrated study, improving our knowledge on the details of cyt c interactions with the supercomplex. However, this seems to be a rather incremental advance, not a major breakthrough in the area.

Reviewer #3

The manuscript by Lobez et al. is an interesting work that brings noteworthy results on the functional role of the mitochondrial respiratory supercomplexes by delving into the diffusion mechanism of cytochrome c (cyt. c) between Complex_III and Complex_IV within the yeast III+IV supercomplex of *S. cerevisiae*.

Notably, the authors provide combined cryo-EM and kinetic data supporting the conclusion that the III+IV supercomplex system is highly responsive to the electrostatic environment and that lowering the ionic strength can reveal a substrate channeling mechanism involving more than one tightly bound cyt. c molecule.

The topic is of great interest for the scientific community who is still largely debating the kinetic advantage of substrate channeling in supercomplexes. Indeed, it is ascertained in the literature that profound differences exist in different organisms, so it would be even more interesting if the authors could further discuss their results in consideration also of recent publications about cyt. c channeling in mammalian mitochondrial respirasomes.

We agree that it would have been interesting to discuss cyt. c channeling in mammalian respirasomes given the structural differences between the different systems. However, the current study is focused on addressing the effect of ionic strength on the structure and function of a specific supercomplex-cyt. c system. Furthermore, to our knowledge, kinetic studies of cyt. c channeling in mammalian supercomplexes/respirasomes have not been presented in the literature. Even so, binding of mammalian cyt. c to mammalian CIII₂/CIV and its relevance to cyt. c channelling in supercomplexes have been discussed. To address the reviewer's comment, we have modified the following sentence (modified or added text in green):

"Binding of cyt. c at two positions at each of a monomer of CIII₂ and CIV at low ionic strength was observed previously and suggested to be involved in electron transfer by cyt. c 2D diffusion also within the mammalian (ref) and plant (ref) supercomplexes."

As minor revisions before publication, we recommend:

- expressing all cyt. c concentrations as "μM" values throughout the text and in the figures, in order to make the comparison between values more immediate;

We have modified the text as suggested by the reviewer, but only kept "nM" at one instance where we refer to a concentration used in another study: "5-10 nM cyt. c".

- changing a typo in the "Isolation and activity of the supercomplex" section, last line: specify "150 mM KCl" in "...fit with a Hill equation to the data at 150 mM" if deemed appropriate.

The typo has been corrected.

Reviewer #4

In the paper entitled “Electron transfer in the respiratory chain at low salinity,” the authors explored the role of salinity in the electrostatic interaction between CIII-CIV supercomplexes and cytochrome c during electron flux. The work is exciting and well-conducted. This reviewer has some questions.

1) *Note that we have switched the order of the two paragraphs below in the reviewer's comments (in black) to answer the more general question first and then address the more specific comment.*

In the text, the authors assume that the cytoplasmic amount of free anions (i.e. Cl) is lower than free cations (i.e. K). In the cytoplasm, all negative charges (protein included) are spontaneously associated with positive charges, and the free ions concentrations must be very low. Please explain why assume that free positive charges are higher than negative charges in the cytoplasm without energy expenditure.

*According to Wennerström *et al.*, in the native system, a majority of negative charges is part of macromolecules (e.g. proteins, DNA, membrane surfaces...). The total amount of charges is ~150-200 mM. However, because a major fraction of anions is part of large polymers, the concentration of small anions is <<150-200 mM. Wennerström *et al.* estimated that under these conditions the Debye screening length is much larger than that of a 150-200 mM monovalent salt concentration and estimated that a realistic mimic of the intracellular Debye screening length is equivalent to 20 mM monovalent salt. This is the reason we used that salt concentration in the current study. In other words, we are not assuming that the concentrations of positive and negative charges are different.*

The authors recover the idea that the cellular concentration of chlorine is low compared to that of potassium (Wennerström, H., *et al.* Proc. Natl. Acad. Sci. USA (2020): 117, 10113-10121); however, the total amount of positive and negative charge is equal in intracellular fluid, in line with the principle of electro-neutrality. In this sense, the amount of Na (8 mEq/L) + K (148 mEq/L) + Mg (35 mEq/L) + Others (7 mEq/L) = Cl (17 mEq/L) + HCO₃ (9.3 mEq/L) + Protein (35 mEq/L) + Phosphate and Organic Acid (137 mEq/L). In essence, all charges (positive or negative) have their countercharge.

*We agree with this analysis, but in Wennerström *et al.* (PNAS (2020): 117, 10113) it is based on the intracellular situation in *E. coli*. Therefore, the actual numbers most likely do not apply to mitochondria. In mitochondrial cristae (where the studied reaction takes place), the (lipid surface):(bulk solution) ratio is large, which means that there is also a considerable contribution to fixed negative charge from a negatively charged membrane surface. We don't know the actual salt concentration that would mimic the Debye screening length in the mitochondrial cristae, but in analogy with the analysis of Wennerström *et al.* it is <<150-200 mM monovalent salt. The purpose of the study is to test how a (more realistic) larger Debye screening length, than that equivalent to 100-150 mM monovalent salt, would influence protein-protein interactions. Another perspective of the study is that that we have introduced an ionic strength perturbation to the system and analysed its functional and structural consequences.*

We would like to thank the reviewer for bringing up the issues discussed in the two above paragraphs as they need some additional clarification. We have addressed the reviewer's

comments by the following changes introduced in the revised version of the manuscript (modified or added text in green):

"A vast majority of the colloidal-sized macromolecules such as nucleic acids, membranes and proteins carry a net-negative charge (refs). In addition, charged organic compounds are typically anions (e.g. ADP/ATP, glutamate, acetate). In other words, to maintain electroneutrality in the intracellular medium, the concentration of small anions such as Cl^- is much lower than the concentration of cations such as K^+ ."

"Because the majority of negative charges are associated with large polyanions, electrostatic interactions between macromolecules within the cell are considerably stronger than in a 150 mM salt solution, yielding a Debye screening length (λ_D) of ~ 2.2 nm instead of $\lambda_D \cong 0.8$ nm in 150 mM monovalent salt. As shown by Wennerström *et al.* {Wennerström, 2020 #4685} a realistic mimic of the intracellular electrostatic conditions is ~ 20 mM monovalent salt."

"The recent finding that the use of a 150 mM "physiological" ionic strength ($\lambda_D \cong 0.8$ nm) yields much weaker electrostatic interactions than those encountered within the cell encouraged us to investigate the effect of lowering the ionic strength on the mechanism of electron transfer between CIII₂ and CIV. The study of Wennerström *et al.* {Wennerström, 2020 #4685} is based on a quantitative analysis of *E. coli* cells. The charge distribution in the mitochondrial intermembrane space is likely to be different from that in *E. coli*, but the same principles governing the Debye screening length apply. To investigate the effect of a more realistic, longer Debye screening length, we measured the kinetics of electron transfer from QH₂ to O₂ as a function of cyt. *c* concentration and determined the cryo-EM structure of the CIII₂CIV_{1/2}-cyt. *c* co-complex at 20 mM monovalent salt concentration ($\lambda_D \cong 2.2$ nm)."

2) Authors showed that the activity of complex III is similar in the presence of 20 mM KCl + 50 μM cyt *c* (80 ± 6 e/s) or 150 mM KCl + 50 μM cyt *c* (90 ± 20 e/s); while the activity of complex IV from supercomplexes rise when KCl increased in the presence of ascorbate plus TMPD: 20 mM KCl and cyt *c* (120 ± 10 e/s) and 150 mM KCl (450 ± 20 e/s). The data showed that the rate-limiting step in the electron flux is complex III. Similar to the metabolic flux analysis showed.

Then, in Fig. 2 the authors suggested that the reduction of salinity increased the affinity of the supercomplex III/IV for the cytochrome *c* (from 1.7 μM to 180 nM). Complex III increases its affinity for cyt *c* oxidized, or complex IV increases its affinity for cyt *c* reduced? Please explain.

We agree with the reviewer's analysis, but would like to add some clarification. Measurements of the activity of each complex (III or IV) were performed at saturating cyt. *c* concentration of 50 μM , which was chosen to obtain a maximum rate (c.f. catalytic constant of each complex). The data in Fig. 2 show a cyt. *c* titration under conditions where electron transfer from CIII to CIV is rate limiting. It does not reflect the affinity for oxidized cyt. *c* to CIII nor that of reduced cyt. *c* to CIV, but rather 2D diffusion of cyt. *c* along the CIII-CIV supercomplex surface. In other words, at steady state, a fraction of cyt. *c* would be reduced or oxidized depending on the relative rates by which cyt. *c* is reduced and re-oxidized, respectively. We have modified and added the following text to the revised version of the manuscript:

"This difference is presumably due to a tighter electrostatic association of cyt. *c* with the supercomplex surface at the lower ionic strength. At low cyt. *c*:CIV ratios electron transfer between each monomer of CIII₂ and CIV presumably takes place by 2D diffusion, which would be slowed by a tighter association of both reduced and oxidized cyt. *c* with the supercomplex surface."

In this sense, kinetic analysis indicates that at low salinity the $n = 5$, suggests kinetic cooperativity or, as the authors said, the electron flux involves two or more cyt *c* molecules. If cyt *c* reduced is retained next to the active site, increasing enzyme affinity is not involved. Please explain."

One possible scenario is that one cyt. *c* is retained at each of CIII and CIV and that the two cyt. *c* molecules exchange electrons with each other by 2D diffusion. Another scenario is that, while the two cyt. *c*s are retained at CIII and CIV respectively, a third cyt. *c* binds to provide an electron-transfer link between CIII and CIV. We have attempted to clarify this question in the text:

"Hence, we propose that the transition in the range 0.1 μM - 0.3 μM cyt. *c* involves two bound cyt. *c* molecules (scenario (i)), one at a CIII₂ monomer and a second cyt. *c* at the adjacent CIV. Electron transfer between the CIII₂ monomer and CIV would take place by 2D diffusion of the two cyt. *c* molecules such that they would exchange electrons upon interaction at the surface."

The equation presented by the authors in the manuscript can be modified as the equation for an essential enzyme activator. Can the reduced cyt *c* act as an activator of the supercomplex?

It is true that the equation could be modified as suggested, but we do not have any data suggesting that binding of cyt. *c* would activate the supercomplex. Because our current conclusion is independently supported by the cryo-EM data, we interpret the data in terms of cyt. *c*(s) that shuttle electrons between complexes III and IV within the supercomplex and that the supercomplex activity increases with an increasing number of bound cyt. *c* molecules.

In Fig. S1 the authors showed the QH₂ oxidation-O₂ reduction rate over time (140 nM/s, as calculated from the oxygen consumption record). Please include the oximeter chamber volume to determine the quinol:oxygen stoichiometry (mol/mol) and compare the activity of the supercomplex and the activity of individual complexes. Do these activities agree with the electron flux?

The oxygraph chamber volume is 1 ml. The information has been added to the materials and methods section. As pointed out above, activities of CIII₂ and CIV are the maximum activities at saturating cyt. *c* concentration. Each of these activities is larger than that of the supercomplex, i.e. all activity values are consistent with the overall electron flux. To answer the reviewer's question, we added the following text to the revised version of the manuscript:

" At cyt. *c* concentrations above $\sim 0.15 \mu\text{M}$, the rate increased sharply to reach saturation at ~ 50 electrons/s at the highest cyt. *c* concentration ($\sim 50 \mu\text{M}$), consistent with rate limiting cyt. *c*-mediated electron transfer between CIII₂ and CIV."

3) In Fig. 3, authors showed that cyt c reduced increase vs cyt c addition, in the presence of 20 mM KCl. Simultaneously, CIV activity was decreased with 20 mM KCl, in contrast to its high activity with 150 mM KCl. The accumulation of cyt c reduced could be the effect of low salinity on the CIV? Please explain.

This is an interesting point and the effect could, in principle, play a role. However, a comparison is not straightforward because the activity of CIV is the maximum activity of this complex (c.f. k_{cat}), while the fraction reduced cyt. c (c.f. Fig. 3) is measured under conditions when electron transfer from CIII to CIV, via cyt. c, is rate limiting. To clarify this issue, we have added the following text to the revised version:

"In principle, a larger fraction or concentration of reduced cyt. c at 20 mM than at 150 mM KCl could be explained by a lower activity of CIV at 20 mM than at 150 mM KCl. However, the CIV activity is presumably not rate limiting because the maximum O_2 -reduction activity of CIV is much larger than $QH_2:O_2$ oxidoreductase activity of the supercomplex. Instead, we speculate that..."

Finally. Isolated supercomplexes showed $QH_2:O$ oxidoreductase activity without cyt c addition? If did, this means that cyt c is tightly bound to the supercomplex, as reported previously by Reyes-Galindo et al. (B.B.A-Bioenergetics. 2019; 1860: 618–627) and can be a reinforcement of the hypothesis.

The activity at [cyt.c]=0 is 0 in the purified sample because cyt. c is removed during purification of the supercomplex. The logarithmic scale is a reason why this is not shown in the figure, but it is mentioned in the text of the revised version. To further discuss this topic, we have added the following text:

" Diffusion of cyt. c between each monomer of $CIII_2$ and CIV within the $CIII_2CIV_{1/2}$ supercomplex is supported by electrostatic interactions between the positively charged docking surface of cyt. c and the negatively charged supercomplex surface. These interactions are manifested by an association of cyt. c with supercomplexes isolated from e.g. *Ustilago maydis* (reference added) or *S. cerevisiae* (references added)."

Reviewers' Comments:

Reviewer #4:

Remarks to the Author:

I have read the authors' responses to this reviewer's questions and I agree with the new version of the manuscript and suggest approving its publication in the journal.